# Dynamics of Magnetic Fluids in Crossed DC and AC Magnetic Fields

**DOI:** 10.3390/nano9121711

**Published:** 2019-11-30

**Authors:** Alexander Pshenichnikov, Alexander Lebedev, Alexey O. Ivanov

**Affiliations:** 1Institute of Continuous Media Mechanics UB RAS, 1, Korolyov str., Perm 614013, Russia; lav@icmm.ru; 2Faculty of Physics, Perm State University, Bukireva St. 15, Perm 614990, Russia; 3Ural Federal University, Lenin av. 51, Ekaterinburg 620000, Russia

**Keywords:** magnetic fluid, crossed magnetic fields, magnetite nanoparticles, relaxation times, dynamic susceptibility, interparticle interactions, chains, clusters

## Abstract

In this study, we derived the equations describing the dynamics of a magnetic fluid in crossed magnetic fields (bias and alternating probe fields), considering the field dependence of the relaxation times, interparticle interactions, and demagnetizing field has been derived. For a monodisperse fluid, the dependence of the output signal on the bias field and the probe field frequency was constructed. Experimental studies were conducted in a frequency range up to 80 kHz for two samples of fluids based on magnetite nanoparticles and kerosene. The first sample had a narrow particle size distribution, low-energy magneto dipole interactions, and weak dispersion of dynamic susceptibility. The second sample had a broad particle size distribution, high-energy magneto dipole interactions, and strong dispersion of dynamic susceptibility. In the first case, the bias field led to the appearance of short chains. In the second case, we found quasi-spherical clusters with a characteristic size of 100 nm. The strong dependence of the output signal on the particle size allowed us to use the crossed field method to independently estimate the maximum diameter of the magnetic core of particles.

## 1. Introduction

The behavior of magnetic fluids in crossed magnetic fields (direct current (DC) bias *H*_0_ and alternating *h = a* cos *ωt*) features a number of peculiarities, which provide valuable information on the internal structure of colloidal solutions, including the characteristic sizes of particles and clusters. The source of information is the dependence of the electromotive force (emf) *E(H*_0_), induced in the measuring coil at double frequency (output signal) on the bias field strength *H*_0_. The specific feature of the experiment is that the output signal changes nonmonotonically with increasing *H*_0_. Similar studies were conducted previously [1,2,3] in the low-frequency region corresponding to a quasistatic limit *ωτ* << 1, where *τ* is the magnetization relaxation time. In this paper, we derive equations that are valid at any value of *ωτ*, provided that the frequency of the probe field *ω* remains small compared with the frequency of ferromagnetic resonance at 10^10^ Hz [4,5]. These equations are used to analyze the experimental data over a wide frequency range. We focus on the polydispersity of particles and interparticle interactions causing particle aggregation. The influence of the magnetic dipole–dipole interparticle interactions is most dramatic in weak and moderate fields. Depending on the particle concentration, the interparticle interactions may be responsible for the two- or four-fold growth in the initial magnetic susceptibility and increase in the nonlinearity of the magnetization curve [6,7]. The polydispersity of particles (i.e., broad particle-size distribution) affects practically all physical properties of magnetic fluids. Consideration of polydispersity can result in a qualitatively new interpretation of experimental data. In this work, we consider the crossed-field method as the tool for obtaining information about the biggest particles contained in a ferrofluid. The coarse fraction leads to the formation of nanoscale (tens of nanometers [8,9,10]) and drop-like (microns and tens of microns [11,12,13,14]) aggregates, and fluid separation into weakly and strongly concentrated phases.

## 2. Methods and Materials

We considered a sample of magnetic fluid, which had the form of a long cylinder, whose axis was aligned with the applied bias field of the intensity *H*_0_ (Figure 1A,B). The weak alternating field *h_0_*(*t*) = *a_0_* cos*ωt* was directed normally to the cylinder axis. The measuring coil enclosed the middle part of the sample and its axis coincided with that of the sample. Simultaneous application of the bias and probing alternating fields caused the vector of the total field **H** to oscillate in the vertical plane. The magnetization **M** executed the same oscillations. Although the field projection on the z-axis was constant, the corresponding magnetization projection *M*_z_ oscillated with time at double the frequency due to the nonlinear dependence *M*(*H*). These oscillations of *M*_z_ induced the output signal. According to the Faraday law, the value of emf in the measuring coil is given by:
(1)$$E=\mu_{0}NS\frac{\partial M_{z}}{\partial t},$$
where *µ_0_* = 4π × 10^–7^ H/m, *S* is the area of the sample cross-section, and *N* is the number of turns in the measuring coil. Pshenichnikov et al. [3], in the case of a weak probe field and low frequency (*ωτ* << 1), showed that the output signal is described by:
(2)$$E(t)=\frac{2\mu_{0}NS\omega a_{0}^{2}H_{0}}{{\left(2H_{0}+M_{0}\right)}^{2}}\left(\frac{M_{0}}{H_{0}}-\frac{\partial M_{0}}{\partial H_{0}}\right)\sin2\omega t,$$
where $M_{0}=M(H_{0})$. Equation (2) is applicable to any magnetic fluid, including concentrated polydisperse solutions with strong interparticle interactions.

The case using diluted solutions was of interest to the analysis because it can demonstrate the influence of particle polydispersity on the magnitude of the signal. The interparticle interactions and demagnetization fields can be neglected and the magnetization curve is described by the superposition of the Langevin functions:
(3)$$M(H)=n\int_{0}^{\infty }m(x)L(\xi )F(x)dx,\;L(\xi )=\coth\xi -1/\xi ,\;\xi =\mu_{0}mH/kT,$$
where *x* and *m*(x) are the diameter of the magnetic kernel and magnetic moment of the colloidal particle, respectively, *n* is the particle number density, *F*(*x*) is the particle size distribution, *ξ* is the Langevin parameter, and *L*(*ξ*) is the Langevin function. Expanding the Langevin function in terms of the argument at *ξ* << 1, we obtain:
(4)$$E_{0}=\frac{\mu_{0}NSn\omega a^{2}H_{0}}{45}{\left(\frac{\mu_{0}}{kT}\right)}^{3}<m^{4}>.$$

The angle brackets in the right-hand part of Equation (4) mean averaging over the ensemble of particles using the distribution function *F*(*x*). The point deserving particular attention strongly depends on the signal of the particle dimension. Since the shape of the single domain particle is close to that of a sphere, its magnetic moment is proportional to the cubed diameter, *m = π M_s_ x*^3^/6, where *M_s_* is the saturation magnetization of the magnetic kernel. So, the contribution of separate fractions to the output signal is proportional to the magnetic moment of the 4th power or the diameter of the 12th power. Then, the form of the function *F*(x) strongly affects the output signal and the coarse dispersed fractions are the main contributor. Here, saturation magnetization of the magnetic fluids is determined by the average magnetic moment of particles, whereas susceptibility in the Langevin approximation by the mean square of the magnetic moment is:
(5)$$M_{\infty }=\langle m\rangle n,\;\chi_{L}=\frac{\mu_{0}\langle m^{2}\rangle n}{3kT}.$$

The experimental setup is schematically shown in Figure 2. Magnetic fluid was poured into a 160 mm long test tube, with an internal diameter of 8 mm. The bias field *H*_0_ varied from 0 to 25 kA/m. The amplitude of the probe alternating current (AC) field varied depending on the experimental conditions, but in all cases did not exceed 2 kA/m. In the experiments, we studied two samples of magnetic fluids (samples No. 1 and 2) of the magnetite–kerosene–oleic acid type obtained by diluting kerosene with base fluids FM1 and FM2, respectively. The base fluid FM1 was obtained following the standard chemical precipitation method [15] in the Institute of Continuous Media Mechanics UB RAS (Perm, Russia), FM2 fluid was prepared in the Ivanovo State Power University (Ivanovo, Russia). They differed mainly in particle size distributions. The desired particle size distribution was obtained by varying the synthesis conditions (concentration of iron and ammonia salts and pH of the solutions, temperature, solution feed rate, and mixing intensity) [16]. The free oleic acid was removed by replacing the dispersion medium [15].

The volume fraction of the crystalline magnetite in the solution was calculated using the magnetic fluid density *ρ_f_* under the assumption that the density of protective shells inessential differed from the density of kerosene *ρ_k_* = 0.78 g/cm^3^:
$$\varphi_{s}=\frac{\rho_{f}-\rho_{k}}{\rho_{mag}-\rho_{k}},$$
where *ρ_mag_* = 5.24 g/cm^3^ is the bulk magnetite density. The use of a more accurate formula for *φ_s_* was unreasonable due to the lack of reliable information on the effective density of the protective shell.

Magnetization curves were determined using the sweep method, in which the differential magnetic susceptibility *χ*(*H*) = *dM/dH* of the fluid was directly measured and the magnetization curve was found by the numerical integration [17]:
(6)$$M(H)=\int_{0}^{H}\chi (H)dH.$$

A long-cooled solenoid with two galvanic isolated coaxial coils was used as the source of the magnetic field. Direct current (DC) was passed through one of the coils, and the weak AC with an infra-low frequency of 0.1 Hz was passed through the second coil. The frequency was sufficiently low, which allowed us to exclude the relaxation processes. The design of the experimental setup enabled the measurement of the amplitudes of small magnetization oscillations and the field strength. The ratio of these quantities provided the desired differential susceptibility value.

The particle size distribution was determined during the magneto-granulometric analysis, described in Pshenichnikov et al. [17]. The analysis allowed us to determine the particle numerical concentration, the average magnetic moment, and its average squared value without any assumptions about the size distribution of particles. The information about the particle diameters was obtained using the two-parameter Γ-distribution with the assumption that the shape of the particles was close to spherical. The Γ-distribution has the form of an asymmetric bell and is described by:
(7)$$f(x)=\frac{x^{\alpha }exp(-x/x_{0})}{x_{0}^{\alpha +1}\Gamma (\alpha +1)},\;\langle x^{q}\rangle =\int_{0}^{\infty }x^{q}f(x)dx=x_{0}^{q}(\alpha +1)(\alpha +2)\ldots (\alpha +q),$$
where Γ(*α* + 1) is the Γ-function, *x*_0_ and *α* are the distribution parameters, and 〈*x^q^*〉 is the moment *x* of order *q*. In particular, the average core diameter 〈*x*〉 and the relative distribution width *δ_x_* are described by the following equations:
(8)$$\langle x\rangle =x_{0}(\alpha +1),\;\delta_{x}=\sqrt{\frac{\langle x^{2}\rangle }{{\langle x\rangle }^{2}}-1}=\frac{1}{\sqrt{1+\alpha }}.$$

The basic parameters (temperature at which the magnetization curve was measured, saturation magnetization *M*_∞_, initial susceptibility *χ*_0_, average magnetic moment <*m*>, mean square magnetic moment <*m*^2^>, average diameter of the magnetic core of particles <*x*>, volume fraction *φ*_ρ_ of magnetite, and relative distribution width *δ_x_*) are presented in Table 1 for the base magnetic fluids.

The long tail of the distribution function in Equation (7) implies that the colloidal solution contains a number of large particles with high energy magneto dipole interactions, which significantly affect the physical properties of magnetic fluids. Primarily, this effect manifests itself in the low magnetic fields. It is described with good accuracy by the modified second-order mean field model [6]. According to this model, the initial equilibrium susceptibility *χ* of the magnetic fluid can be represented as a series expansion in powers of the Langevin susceptibility *χ_L_*:
(9)$$\chi_{0}=\chi_{L}\left(1+\frac{\chi_{L}}{3}+\frac{\chi_{L}^{2}}{144}\right),\;\chi_{L}=\frac{\mu_{0}m^{2}n}{3kT}.$$

A comparison of the initial susceptibility of the base fluids (Table 1) with the Langevin susceptibility calculated by Equation (9) showed that the magneto dipole interactions led to an almost two-fold increase in the initial susceptibility. The magnetic properties of diluted samples directly used in the experiments are shown in Table 2. The volume fraction of the magnetic phase in the samples (defined as the ratio of the saturation magnetization of the sample to the saturation magnetization of the magnetite *M_s_* = 480 kA/m) did not exceed 1.7%, which did not diminish the marked contribution of the magneto dipole interactions. The susceptibility due to the interactions increased from 7% for sample No. 1 to 27% for sample No. 2, containing the maximum fraction of coarse particles.

The dynamic susceptibility measurements were performed using the mutual induction bridge (MIB) described previously [18,19]. The complex susceptibility $\hat{\chi }=\chi^{\prime }-i\chi^{\prime \prime }$ is related to the output voltages (differential Δ*U* and reference *U*_4_ voltages) by the simple equation:
(10)$$\hat{\chi }=\frac{S_{0}\Delta \hat{U}}{S{\hat{U}}_{4}},$$
where *S*_0_ and *S* are the cross-sectional areas of the coil and the sample, respectively. Equation (10) allowed us to calculate the desired susceptibility components in terms of voltages Δ*U* and *U*_4_ and the phase shift between them. The amplitudes and phases of the two voltages were measured with a dual-channel synchronous amplifier eLockIn 203 (Anfatec Instruments AG, Oelsnitz Saxony, Germany). For the experiment conditions (the sample in the form of the long cylinder), the demagnetizing factor of the sample was sufficiently small (κ = 0.0065 ± 0.0005) and was used to compute the correction for the voltage enabled determination of the maximum error of measurement of susceptibility *χ*′, which is equal to $\pm (0.2+2\chi )\times 10^{-2}$. The measurement error for *χ*″ was at the level of 0.01 SI units for diluted magnetic fluids and did not exceed 5% of the static susceptibility value for concentrated solutions. The coupling constant *λ* in Table 2 is the ratio of dipole–dipole interaction energy to thermal energy and is discussed in detail in Section 3.3.

## 3. Results

### 3.1. Relaxation Processes

We determined the output signal at an arbitrary frequency of the probe field. In this case, the magnetization of the sample differed from the equilibrium value, the co-linearity of the vectors **M** and **H** was violated (Figure 1B), and Equation (2) became inapplicable. To determine the components of magnetization, it was necessary to solve the relaxation equation, which contains the characteristic relaxation times *τ* of the magnetic moment taken as parameters. Real magnetic fluids, as a rule, have broad particle-size distributions and wide ranges of magnetization relaxation times. Considering the polydispersity of particles in the dynamic problem makes it unnecessarily cumbersome. Therefore, in this section, we restrict our discussion to the monodisperse ferrofluids, in which all particles are identical.

Since, according to the experimental conditions, the AC field component was small, we used the linearized relaxation equation [20,21]
(11)$$\frac{dM_{x,z}}{dt}=-\frac{1}{\tau_{⊥,∥}}\;{\left[M-\bar{M}(H)\right]}_{x,z},$$
where $\bar{M}(H)$ is the equilibrium magnetization corresponding to the instantaneous value of the field strength in the sample. The relaxation times *τ*_II_ and $\tau_{⊥}$ for the longitudinal and transverse components of magnetization, respectively, depend on the field strength; in the dilute solution they are described by [20,21]:(12)$$\tau_{∥}=\frac{\xi L^{\prime }(\xi )}{L(\xi )}\;\tau_{B},\;\tau_{⊥}=\frac{2L(\xi )}{\xi -L(\xi )}\;\tau_{B},\;\tau_{B}=3\eta V/kT,$$
where *τ*_B_ is the Brownian time of rotational diffusion of particles in zero magnetic field, *η* is the ferrofluid viscosity, and *V* is the volume of the particle covered with a protective shell. In weak fields (*ξ* << 1), both relaxation times coincide with the Brownian time *τ*_B_ and monotonically decrease with the growth of the magnetic field.

Expanding the equilibrium magnetization,
$$\bar{M}(H)=\left(\frac{M}{H}h,\;\frac{M}{H}H_{0}\right),$$
in Equation (11) in a power series of the weak ac field *h*, and retaining only the terms of the lowest power, we obtain:
$$\frac{\partial M_{x}}{\partial t}=-\frac{1}{\tau_{⊥}}\left[M_{x}-\frac{M(H_{0})}{H_{0}}\;h\right],$$
(13)$$\frac{\partial M_{z}}{\partial t}=\frac{1}{\tau_{∥}}\left[M_{z}-M(H_{0})-\frac{h^{2}}{2H_{0}}{\left(\frac{\partial M}{\partial H}-\frac{M}{H}\right)}_{H=H_{0}}\right].$$

Then, substituting the explicit expression for the internal field *h* = *a* cos *ωt* in Equation (13), we find:
(14)$$M_{x}=\frac{M(H_{0})a}{H_{0}\sqrt{1+\omega^{2}\tau_{⊥}^{2}}}\;\cos(\omega t-\beta_{2}),$$
(15)$$M_{z}=M(H_{0})\;+\;\frac{a^{2}}{4H_{0}}{\left(\frac{\partial M}{\partial H}-\frac{M}{H}\right)}_{H=H_{0}}\left[1+\frac{\cos(2\omega t-\beta_{1})}{\sqrt{1+4\omega^{2}\tau_{∥}^{2}}}\right],$$
$$\mathrm{tg}\;\beta_{1}=2\omega \tau_{∥},\mathrm{tg}\;\beta_{2}=\omega \tau_{⊥}.$$

Using Equation (14) and the relationship between the applied and internal probe fields given as:
$$h=h_{0}-\kappa_{x}M_{x},$$
we find the amplitude of the probe field inside the sample:
(16)$$a^{2}=\frac{a_{0}^{2}(1+\omega^{2}\tau_{⊥}^{2})}{{\left[1+\frac{M(H_{0})}{2H_{0}}\right]}^{2}+\omega^{2}\tau_{⊥}^{2}}.$$

Substituting Equations (15) and (16) into Equation (1) yields the required expression for the output signal:
(17)$$E_{0}=\frac{2\mu_{0}NS\omega a_{0}^{2}H_{0}(1+2\omega^{2}\tau_{⊥}^{2})}{\sqrt{1+4\omega^{2}\tau_{∥}^{2}}\left[{\left(2H_{0}+M_{0}\right)}^{2}+4H_{0}^{2}\omega^{2}\tau_{⊥}^{2}\right]}\left(\frac{M_{0}}{H_{0}}-\frac{\partial M_{0}}{\partial H_{0}}\right).$$

Equation (17) determines the field and frequency dependences of output signal. It is valid for an arbitrary magnetic fluid with a rather narrow particle size distribution and negligible spread of particles in the relaxation time.

To analyze the experimental data, we introduced the normalized output signal *E** by dividing Equation (17) into a coefficient that included the parameters of the measuring coil, the frequency, and square of the probing field amplitude. The normalized signal thus describes only the effect of the bias field and relaxation processes:
(18)$$E^{*}=\frac{E_{0}}{2\mu_{0}NS\omega a_{0}^{2}}=\frac{\left(1+2\omega^{2}\tau_{⊥}^{2}\right)}{\sqrt{1+4\omega^{2}\tau_{∥}^{2}}\left[{\left(2+M_{0}/H_{0}\right)}^{2}+4\omega^{2}\tau_{⊥}^{2}\right]H_{0}}\left(\frac{M_{0}}{H_{0}}-\frac{\partial M_{0}}{\partial H_{0}}\right).$$

We theoretically analyzed Equation (18), considering the steric and magneto dipole–interparticle interactions. For this purpose, we used the modified effective field model [6], which adequately describes the equilibrium magnetization of concentrated ferrofluids. Samples studied in the experiment (Table 2) had relatively low concentrations of particles; therefore, when calculating the effective field, we considered only the correction, which was linear in concentration (i.e., proportional to Langevin susceptibility):
(19)$$M_{0}=mnL(\xi_{e}),\;\xi_{e}=\xi_{0}+\chi_{L}L(\xi_{0}),$$
where $\xi_{0}=\mu_{0}mH_{0}/kT$ is the Langevin parameter, which is determined in terms of the bias field strength. Substituting Equation (19) into (18) leads to the expression:
(20)$$E^{*}=\frac{\mu_{0}m}{kT}\frac{3\chi_{L}(1+2\omega^{2}\tau_{⊥}^{2})}{\sqrt{1+4\omega^{2}\tau_{∥}^{2}}\left[{\left(2+3\chi_{L}L(\xi_{e})/\xi_{0}\right)}^{2}+4\omega^{2}\tau_{⊥}^{2}\right]\xi_{0}}\left[\frac{L(\xi_{e})}{\xi_{0}}-\frac{\partial L(\xi_{e})}{\partial \xi_{0}}\right].$$

The field dependence of the output signal calculated by Equation (20) is shown in Figure 3A,B. To compare the experimental results, the magnetic moments *m* of the particles and the Langevin susceptibility *χ*_L_ of the model fluids were considered identical to those of samples No. 1 and 2 in Table 1 and Table 2. Since Equations (19) and (20) do not consider the polydispersity of particles, one good quantitative agreement between the calculated and experimental data in the case of the broad size distribution of particles should not be expected. The main advantage of Equations (18) and (20) is that they consider the dynamic effects, magneto dipole–interparticle interactions, and the demagnetizing field. The last two factors compete with each other. The magneto dipole interactions increase ferrofluid magnetization, whereas the demagnetizing field causes its decrease. Their total effect remains appreciable even for moderately concentrated solutions, such as samples No. 1 and 2. In Figure 3A,B this effect is manifested in the shift of the maximum of the function *E** = *f*(*ξ*) toward weak fields. For dilute magnetic fluids in which the interparticle interactions and demagnetizing fields are negligible, the Langevin parameter, corresponding to the maximum of the function *E** = *f*(*ξ*), is equal to *ξ* =* 1.93 [2,3]. Equation (2) supports the correctness of this value in the case when the fluid magnetization is calculated in the Langevin approximation. However, the curves in Figure 3 constructed with regard for the interparticle interactions demonstrate markedly lower values of *ξ**: *ξ* ≈* 1.53 for sample No. 1 and *ξ* ≈* 1.65 for sample No. 2.

The shapes of the *E** = *f*(*ξ,ω*) curves in Figure 3A,B for two model monodisperse fluids are qualitatively identical. With increasing frequency of the probe field (*ωτ*_B_ ≥ 1), the output signal decreased and the maximum was smeared. In the strong bias field (*ξ_0_* ≥ 10), relaxation times decreased, as predicted by Equation (12), therefore, the dynamic contributions to Equations (18) and (20) decreased by almost two orders of magnitude. The process of magnetization reversal became quasi-static and the output signal was no longer dependent on the frequency, at least at *ωτ*_B_ ≤ 6.

### 3.2. Results of Dynamic Susceptibility Experiment

The real ferrofluids have a wide spectrum of relaxation times due to the polydispersity of single-domain particles, the formation of the clusters, and two independent relaxation mechanisms (Brownian and Néel). The Brownian mechanism of magnetization relaxation is associated with the rotation of particles in a viscous medium, and the characteristic relaxation time is determined by Equation (12). The Néel mechanism of magnetization relaxation is associated with the rotation of the magnetic moment of the particle relative to the crystallographic axes [22,23,24]. For the uniaxial single-domain particle the most energetically favorable orientations of the magnetic moment are the two opposite directions along the easy axis. These two states are separated by the energy barrier *KV*_m_, where *K* is the magnetic anisotropy constant and *V*_m_ is the volume of the particle magnetic core. In the weak field, this barrier can be overcome due to thermal fluctuations within the particle itself, which corresponds to the Néel relaxation mechanism.

The Néel time *τ_N_* required to overcome the barrier grows exponentially with decreasing temperature, i.e., *τ_N_* ~ *τ*_0_ exp *σ*, where *τ*_0_ ~ 10^−9^ s is the damping time of the Larmor precession, and *σ* is the reduced barrier height (anisotropy parameter):
(21)$$\sigma =\frac{KV_{m}}{kT}.$$

The magnetization dynamics in the weak applied field are determined by that of two relaxation mechanisms, which ensures the shortest relaxation time. The Brownian and Néel relaxation times depend differently on the particle volume [23]. The condition *τ_N_* = *τ_B_* provides the characteristic magnetic core diameter *x*^*^, which corresponds to “switching” off the relaxation mechanism. If *x* < *x**, then the Néel relaxation mechanism prevails, and if *x* > *x*^*^, then the Brownian mechanism is predominant. Generally, *x*^*^ does not coincide with the limiting size of superparamagnetic particles: the Brownian fraction includes both magnetically hard particles and some superparamagnetic particles with *τ_N_* > *τ_B_*. According to estimates [25], for low-viscous magnetite ferrofluids *x*^*^ ≈ 16−18 nm, *τ_N_* = 10^−10^ −10^−5^ s, and *τ_B_* = 10^−5^ −10^−3^ s. Thus, at frequencies up to 10^4^ Hz, the dispersion of the dynamic susceptibility is specified by the particles with Brownian relaxation mechanism and at frequencies above 10^5^ Hz by the particles with the Néel magnetic moment relaxation mechanism.

The frequency dependence of the initial susceptibility for samples No. 1 and 2 with the lowest and highest concentrations of large particles is shown in Figure 4. The *χ*(*ω*) curves differ drastically. For sample No. 1, a weak susceptibility dispersion was observed only at frequencies of the order of 10^5^ Hz; for sample No. 2, the maximum dispersion was observed at frequencies of 300–400 Hz. Such difference in the dynamics of magnetization is the direct result of differences in the particle size distributions.

Sample No. 1 had a relatively narrow particle size distribution. The main contribution to the dynamic susceptibility was from superparamagnetic particles with magnetic core diameter *x < x*^*^ ≈ 16 nm and rather short relaxation times (*τ_N_ <* 10^−5^ s). In the examined frequency range, this sample showed a quasi-static behavior, and the region of susceptibility dispersion was beyond the upper boundary of this range. Conversely, sample No. 2 had a broad particle-size distribution (Table 1) with a long tail, so the main contribution to the dynamic susceptibility was due to the Brownian particles with a magnetic core diameter *x > x** and long relaxation times. Notably, the contribution of large particles to the initial susceptibility was disproportionately high; it grew as the squared magnetic moment or as the sixth power of the diameter. The hydrodynamic diameter *d* of the particles, which contributed the most to the susceptibility dispersion, could be estimated from the condition 2π*f*τ*_B_ = 1, where *f** is the frequency corresponding to the maximum on the *χ*″(*ω*) curve. Substituting *f* ≈* 330 Hz into this condition and the Brownian relaxation time from Equation (12) yielded *d ≈* 100 nm. This hydrodynamic diameter value is approximately three or four times greater than the maximum possible diameter of the individual particles, which in magnetite ferrofluids is determined with an electron microscope. The existence of the surfactant protective shell with a characteristic thickness slightly higher than 2 nm cannot account for such a large difference in size. This led us to conclude that the multi-particle aggregates (clusters) rather than single particles acted as independent kinetic units, which generated the spectrum of dynamic susceptibility of sample No. 2. This conclusion agrees well with the data, which we obtained earlier in similar experiments [25] and in experiments on the diffusion and magnetophoresis of particles in magnetic fluids containing coarse particles [8,26]. The same results were reported [9] using the dynamic light scattering method.

### 3.3. Crossed Field Experiment

The results of the crossed fields experiments are shown in Figure 5A,B in the dimensional coordinates. For both samples, we used the same set of frequencies at which measurements were made. The experimental *E**(*H*) curves for different samples differ significantly. First, at the low-frequency limit, there is an eight-fold difference in the H* positions of maxima of the *E**(*H*) functions (8.0 kA/m and 1.0 kA/m for No. 1 and No. 2, respectively), and the corresponding signal amplitudes differ approximately by a factor of 14. Second, the frequency dependence of the signal for sample No. 1 became distinguishable at frequencies of about 6 kHz, and for sample No. 2 at 60 Hz. Third, with an increase in the frequency of the probe field, the maximum of the function *E**(*H*) for sample No. 1 shifted toward weak fields, and for sample No. 2, it shifted toward strong fields. The curves plotted in Figure 2 predict a discrepancy an order of magnitude weaker between the samples. Since Equation (20) considers all significant factors except for the polydispersity of particles, we can assume that a broad particle size distribution (most pronounced in sample No. 2) was the main cause of the observed discrepancies. The coarse particle fractions existing in magnetic fluids disproportionately contributed to the output signal; therefore, the substitution of the mean value of the magnetic moment from Table 2 into Equation (21) led to a systematic error, which increased with the width of the particle size distribution.

Let us consider the problem of particle size distribution in magnetic fluids in more detail. This deserves special attention, since the size distribution of particles often plays an important role in the comparison of experimental and theoretical results. Thus, for example, when constructing an equilibrium magnetization curve, the third- and sixth order-moments should be used, i.e., <*x*^3^> and <*x*^6^> [17]. It may be necessary to use the moments of the ninth order <*x*^9^> when processing experimental data on birefringence in magnetic fluids, because in weak magnetic fields the output signal grows in proportion to the ninth power of the diameter [27,28].

Figure 6A,B presents curves illustrating the contribution of separate fractions to the saturation magnetization of the ferrofluid proportional to <*x*^3^>, Langevin susceptibility (<*x*^6^>), and output signal in the crossed field experiment (<*x*^12^>). The density of the particle size distribution (curve 1) was calculated by Equation (7) for the Γ-distribution. All curves were normalized to unity. The parameters used in calculations were taken from Table 1 and Table 2. The parameters associated with sample No. 1 can be considered typical of magnetite ferrocolloids, including commercial ferrofluids. Sample No. 2 was specifically chosen to demonstrate the effects associated with polydispersity, and had a very broad particle size distribution.

In Figure 6, the vertical dashed line is used to denote the maximum diameter *x*_max_ ≈ 20–25 nm of magnetite particles, which were still observable in an electron microscope in highly stable magnetic fluids and powders obtained by the chemical precipitation method [9,10,29,30,31,32]. In real solutions, particles with magnetic cores of large diameter are absent, since the concentration of iron salts in solutions and the duration of the chemical reaction are limited. Strictly, the Γ-distribution in Equation (7), which suggests the existence of particles with arbitrary large diameters, contradicts the experimental data on the existence of *x*_max_. However, Equation (7) is often used to approximate the size distribution of particles, since the systematic error associated with the tail of the distribution is inessential when calculating the moments of *x* of low orders, for example, <*x*>. Figure 6 shows that the Γ-distribution correctly describes the size distribution of the particles in both samples. Only a negligible fraction of particles had magnetic cores with diameter exceeding *x*_max_.

The evaluation of high-order moments of *x* did not pose serious difficulties in the case when the width of the particle size distribution was rather small (sample No. 1, Figure 6A). A qualitative change in the situation was observed when calculating the high-order moments of *x* for a large width of the particle size distribution (*δ_x_* > 0.4). Figure 6B shows that even when calculating <*x*^6^>, the systematic error associated with the long tail of the distribution reached 40% and became unacceptably large. The results of calculation of <*x*^12^> (curve 4, Figure 6B) are unreliable due to uncertainty about the concentration of large particles (with diameters *x* ≈ 25 nm and higher). Notably, replacing the Γ-distribution with the lognormal distribution, which is often used to analyze experimental data [27,28], does not resolve the issue. The lognormal distribution has a longer tail than the Γ-distribution and the systematic error in calculating high-order moments will be even higher.

We used the *E**(*H*) calculated curves for the model monodisperse liquids plotted in Figure 3 and the experimental curves from Figure 5 to estimate the magnetic moments *m*_e_ of particles in the coarse fraction, responsible for the appearance of the maximum on the *E**(*H*) curve and contributing the most to the normalized signal. To this end, we equated the Langevin parameter *ξ**, corresponding to the maximum of the signal in Figure 3, to the Langevin parameter determined in terms of the effective magnetic moment *m_e_* and the value of the bias field *H** in Figure 5:
(22)$$\xi *=\mu_{0}m_{e}H*/kT,\;m_{e}=\pi M_{s}x_{\max}^{3}/6,$$
where *M*_s_ = 480 kA/m is the saturation magnetization of the magnetite. For sample No. 1 we obtained *ξ** = 1.53 and *m*_e_ = 6.2 × 10^−19^ A·m^2^, which is a value three times higher than the average magnetic moment <m> = 2.08 × 10^−19^ A·m^2^ in Table 1. The maximum diameter of the magnetic core of particles was *x*_max_ = 13.5 nm, and the hydrodynamic diameter of the particles was *d* = 18 nm. The corresponding Brownian relaxation time was $\tau_{B}=3\eta V/kT=2\times 10^{-6}s$. This implies that the dynamic effects in weak fields should be observed at frequencies higher than 80 kHz, which is substantiated by the experimental data on the dynamic susceptibility in Figure 4A. The dispersion of dynamic susceptibility at frequencies up to 80 kHz is rather weak, since particles with the diameter of magnetic cores close to the average value contribute the most to the susceptibility, at which the quasi-static condition *ωτ*_B_ << 1 is fulfilled.

In general, the dynamics of sample No. 1 in the weak bias field (up to 2 kA/m) were consistent with the above statements. In this case, the *E**(*H*) curves had approximately the same slope for all frequencies except for the highest frequency of 80 kHz. The signal dispersion was insignificant due to the absence of large particles in the solution and the short Brownian relaxation time (*ωτ*_B_ << 1). With increasing bias field strength, the relaxation times decrease, according to Equation (12), and the signal dispersion should decrease additionally, as shown in Figure 3A. However, in the experiment, we observed an opposite effect. With the growth of the bias field, the signal dispersion also increased. For *H*_0_ ≥ 5 kA/m the frequency dependence was observed at a frequency of 9 kHz, which implies a three- to four-fold increase in the Brownian relaxation time. In our opinion, this paradoxical behavior of sample No. 1 can only be explained by the formation of short chains in the magnetic fluid at the cost of anisotropic dipole–dipole interparticle interactions and bias field.

As is known [23,33], the probability of the formation of chains in magnetic fluids is determined by the value of the dipolar coupling constant *λ*, which is the ratio of dipole–dipole interaction energy to thermal energy. At *λ* < 1, the effect of aggregates on the properties of magnetic fluids is insignificant, but at *λ* ≥ 2, the number of aggregates increases exponentially. For polydisperse fluid, the value of *λ* can be estimated in terms of the Langevin susceptibility *χ*_L_ and the hydrodynamic concentration *φ* of particles, which were determined from the results of independent experiments: $\lambda =\chi_{L}/\left(8\varphi \right)$ [6,19]. The values of the dipolar coupling constant for the examined samples calculated by this formula are provided in Table 2. For sample No. 1, *λ* = 0.6, and in the weak bias field, the influence of aggregates can be neglected. The application of the stronger field corresponding to the Langevin parameter *ξ* ≥ 1 stimulated the growth of chains and increased the relaxation time of the magnetization due to an increase in the volume of the chain and its form-factor. So, the dispersion of the signal observed in Figure 5A at frequencies higher than 9 kHz is the consequence of the change in the internal structure of the magnetic fluid.

A different situation was observed when analyzing the data for sample No. 2 with the broad particle size distribution in Figure 5B. For this sample, the coupling constant *λ* = 2.1 and the majority of large particles were combined into chains or quasi-spherical clusters already in a zero-bias field. As in the case of linear susceptibility, the strong signal dispersion was already observed at frequencies of 100–300 Hz, which is one more indication of the presence of aggregates with a characteristic size of the order of 100 nm. According to Rosensweig [15], for typical magnetic fluids that are stabilized by oleic acid, the height of the energy barrier associated with the steric repulsion is close to 20 *kT*. This barrier ensures the negligible rate of irreversible particle aggregation and high stability of the magnetic fluid for many years. Experiments devoted to studying the rheology and diffusion of particles in magnetic fluids [26], dynamic magnetic susceptibility [19], and dynamic light scattering method [9] suggest that in the presence of large-sized particles, quasi-spherical aggregates with a characteristic size up to 100 nm have a high probability of occurrence. Our results confirm this inference.

Magneto dipole interactions are not the only reason for the aggregation of particles in ferrofluids. Van der Waals attractive forces and defects in the protective shells [26] can play an important or even a key role in this process. In this case, the application of the bias field did not change the structure of the colloidal solution. The nanoscale aggregates with the uncompensated magnetic moment behaved like single particles. This is why the experimental curves in Figure 5B do not qualitatively differ from the model curves for monodisperse liquid in Figure 2B.

The quantitative discrepancies between the families of curves representing different samples are large and related to the width of the particle size distribution. First, from the maximum condition in Equation (22) for sample No. 2 (*ξ** = 1.65), we obtained an estimate for the effective moment of particles, which contribute the most to the normalized output signal: *m*_e_ = 54 × 10^−19^ A m^2^. This value exceeds the average magnetic moment <*m*> = 2.31 × 10^−19^ A m^2^ already by a factor of 23. The diameter of the magnetic core of such particles should be close to the maximum possible value of *x*_max_. An estimate using Equation (22) provided *x*_max_ ≈ 28 nm, which is only slightly higher than the maximum possible value corresponding to the dashed line in Figure 6, but is significantly smaller than the diameter *x*_max_ ≈ 41 nm obtained with the use of the standard Γ-distribution in Equation (7). This result demonstrates once again the need to replace the Γ-distribution by another distribution characterized by the absence of a long tail.

The easiest way to replace the Γ-distribution is to cut the tail. The distribution in Equation (7) is replaced by the equation:
(23)$$f(x)=\left\{\begin{array}{ll} \frac{Ax^{\alpha }exp(-x/x_{0})}{x_{0}^{\alpha +1}\Gamma (\alpha +1)} & if\;x\leq x_{\max}\\ 0 & if\;x>x_{\max} \end{array}\right\},$$
in which the normalization constant *A* differs from the unit by less than 1%. The truncated Γ- distribution in Equation (23) was used previously [28] for processing the birefringence results and by Aref’ev et al. [34], for calculating high-order moments describing the nonlinear susceptibility of magnetic fluids. In both cases, the use of the truncated Γ-distribution enabled a significant reduction of the discrepancy between the experimental and calculated results. The results reported by Aref’ev et al. [34] revealed a correlation between the numerical value *x*_max_ and the Γ-distribution parameters, which was valid, at least for the magnetite colloids obtained by the chemical precipitation method. According to Aref’ev et al. [34]:
(24)$$x_{\max}=\alpha x_{0}\frac{\langle m^{2}\rangle }{{\langle m\rangle }^{2}}=\alpha x_{0}\frac{\langle x^{6}\rangle }{{\langle x^{3}\rangle }^{2}}.$$

For sample No. 2, the calculations using Equation (30) resulted in *x*_max_ = 29 nm, which practically coincides with the estimate *x*_max_ = 28 nm found by Equation (22). Estimates for sample No. 1 were *x*_max_ = 13.6 and 13.5 nm for Equations (30) and (22), respectively. Thus, the two methods for evaluating the maximum size of particles existing in magnetite colloids agree well for both samples, despite these methods being based on different experimental techniques.

## 4. Discussion and Conclusions

This paper presented the results of experimental and analytical investigation into the dynamics of the magnetic fluid in crossed fields. We examined a situation in which the constant magnetic field *H*_0_ directed along the sample and the weak alternating field normal to its axis act on the sample of magnetic fluid in the form of the long cylinder. The axis of the measuring coil was oriented along the sample and the induction of emf *E*(*H*_0_,ω) was realized at double the frequency. The characteristic feature of the experiments was a strong dependence of the output signal (emf) on the particle size distribution in the weak bias field and the pronounced maximum in the region of the Langevin parameter *ξ* ≈ 1.5–1.9. Our attention was focused on the dependence of the output signal on the probe field frequency and the use of crossed field experiments for obtaining information about the largest particles in magnetic fluids.

We derived equations, describing the dynamics of the magnetic fluid in the crossed magnetic fields, taking into account the magneto dipole interactions, demagnetizing fields, and the field dependence of magnetization relaxation times. For the monodisperse fluid, the dependences of the output signal on the bias field and the probe field frequency were constructed. As can be seen from Equation (20) and Figure 3, in weak and moderate fields, the dynamic effects manifested themselves already at *ωτ*_B_ ≈ 1, but the degree of their influence on the output signal depended on the initial susceptibility of the solution. The higher the susceptibility, the stronger the dynamic effects. With an increase in the bias field, the relaxation times decreased according to formula (12), and at *ξ*_0_ > 10 the dynamic effects became negligible.

The experimental studies were conducted at room temperature in the frequency range from 37 Hz to 80 kHz for two samples of magnetic fluids based on colloidal magnetite and kerosene, which differed in the width of the particle size distribution. The first sample had a narrow particle size distribution, low-energy magnetic dipole interactions, and a low probability of aggregate formation. The weak dispersion of the dynamic susceptibility at the frequencies up to 80 kHz seems to be due to the absence of aggregates. The dynamics of the sample in weak crossed fields (up to 2 kA/m) corresponded to the above statements. The slopes of the *E**(*H*) curves were approximately the same for all frequencies except for the highest frequency of 80 kHz. This result looks natural: the signal dispersion is insignificant due to the absence of large particles in the solution and the small Brownian relaxation time (*ωτ*_B_ << 1). With increasing bias field strength, the relaxation times decreased according to Equation (12), and the signal dispersion should decrease further, as shown in Figure 3A. However, the experiment demonstrated an opposite effect. With increasing bias field, the signal dispersion did not reduce, but increased. Thus, at *H*_0_ ≥ 5 kA/m, the frequency dependence is already fixed at the frequency of 9 kHz, which implies an increase in the Brownian relaxation time by three to four times. In our opinion, such paradoxical behavior of sample No. 1 can only be explained by the formation of short chains in the magnetic fluid due to anisotropic dipole–dipole interparticle interactions and the bias field.

The second sample had a broad particle-size distribution, high-energy magneto dipole interactions, and a strong dispersion of dynamic susceptibility already at frequencies of several hundred hertz (Figure 4). The characteristic sizes of the fluctuating particles estimated according to Equation (12) show that the function of such particles is performed by quasi-spherical clusters with a characteristic size of 100 nm. These sizes agree with the results obtained by other methods reported previously [8,9,10,25,26]. For the second sample in the crossed fields, an increase in the signal amplitude by a factor of 14 could be observed compared to sample No. 1, and a simultaneous eight-fold decrease in the field, corresponding to the maximum on the *E*(*H*) curve.

A strong dependence of the output signal on the particle size allows the crossed field method to be applied for evaluating the maximum diameter *x*_max_ of the magnetic core of particles: *x*_max_ = 13 nm for sample No. 1 and *x*_max_ = 28 nm for sample No. 2. The same values of the maximum diameter (within the experimental error) were obtained in our study using the correlation between *x*_max_ and the moments *x* of the third and sixth orders [34]. These results demonstrate the necessity of re-evaluating the applicability of lognormal and Γ-distributions to computation of high-order moments. In the case of a broad particle size distribution, such calculations are incorrect due to long tails. This problem can be solved by truncation of the tails according to Equation (23). However, this reduction makes simple formulas like Equation (7) inapplicable for arbitrary moments *x*, requiring a numerical calculation of the corresponding integrals.

## Figures and Tables

**Figure 1 nanomaterials-09-01711-f001:**
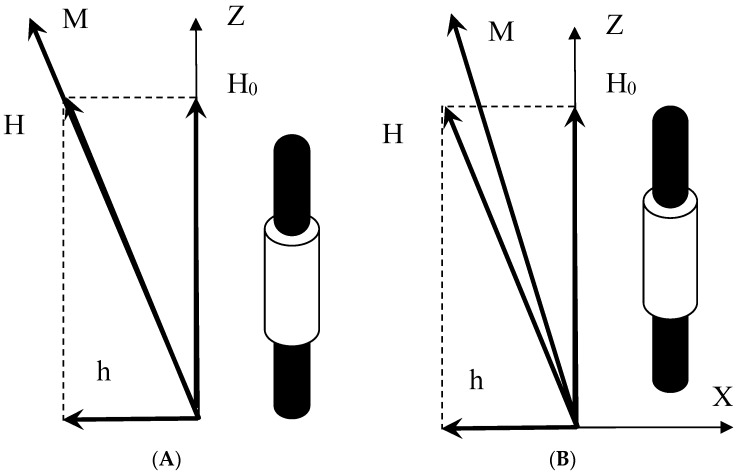
Measuring cell in crossed magnetic fields. The black cylinder is the test tube containing magnetic fluid, the light cylinder is the measuring coil. (**A**) The low frequency limit; (**B**) the arbitrary frequencies.

**Figure 2 nanomaterials-09-01711-f002:**
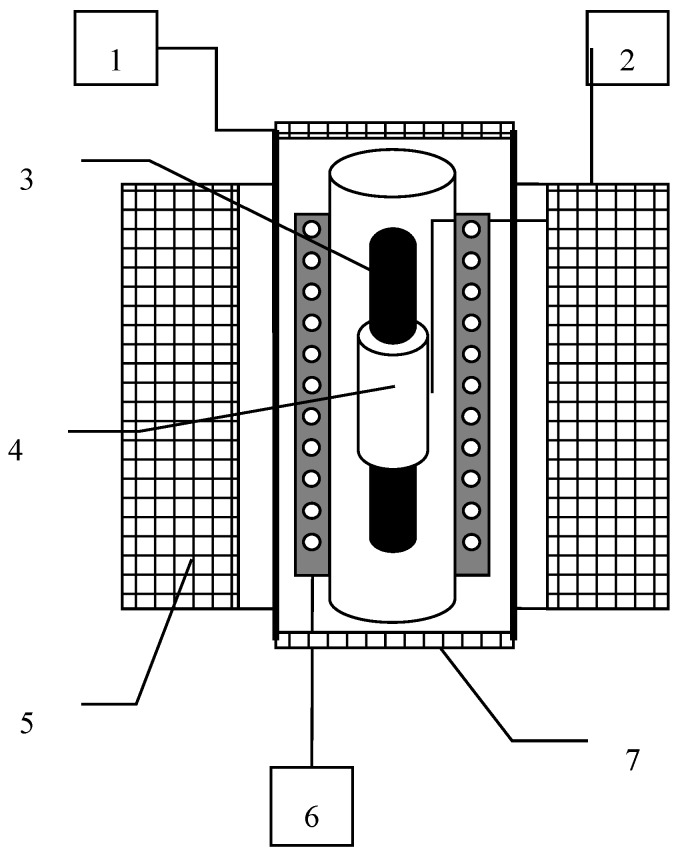
Scheme of the experimental setup. **1**, Sine voltage generator; **2**, temperature sensor; **3**, test tube with magnetic fluid; **4**, measuring coil; **5**, bias coil; **6**, thermostatic shell; **7**, single-layer solenoid for generation of alternating current (AC) field.

**Figure 3 nanomaterials-09-01711-f003:**
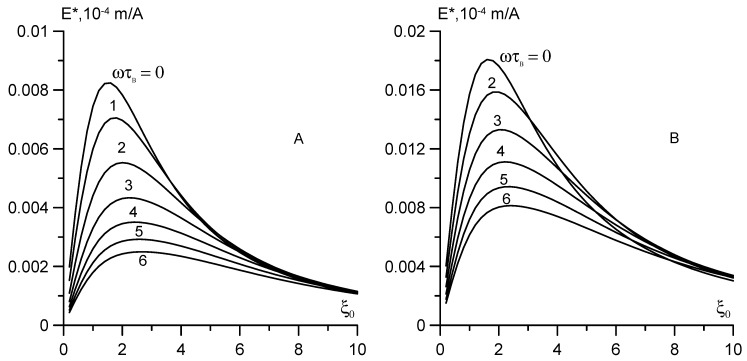
The normalized output signal versus the Langevin parameter for monodisperse ferrofluid and different frequencies of the probe field. (**A**) The simple No. 1 sample (<*m*> = 2.08 × 10^−19^ A m^2^, *χ*_L_ = 0.29; (**B**) the simple No. 2 sample (<*m*> = 2.3 × 10^−19^ A m^2^, *χ*_L_ = 0.84).

**Figure 4 nanomaterials-09-01711-f004:**
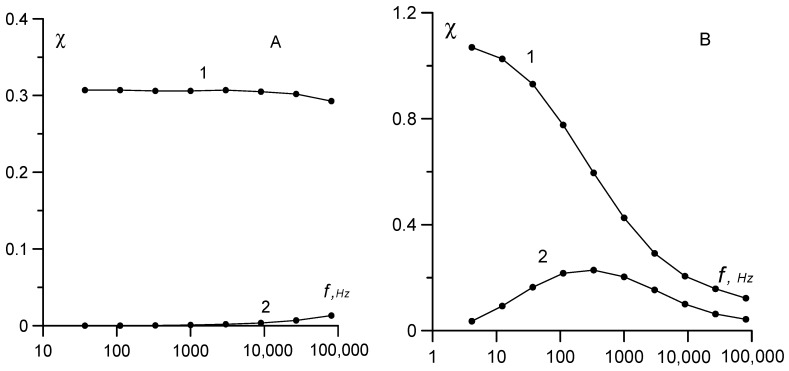
The real (curve 1) and imaginary (curve 2) parts of the dynamic susceptibility versus frequency for (**A**) sample No. 1 and (**B**) sample No. 2.

**Figure 5 nanomaterials-09-01711-f005:**
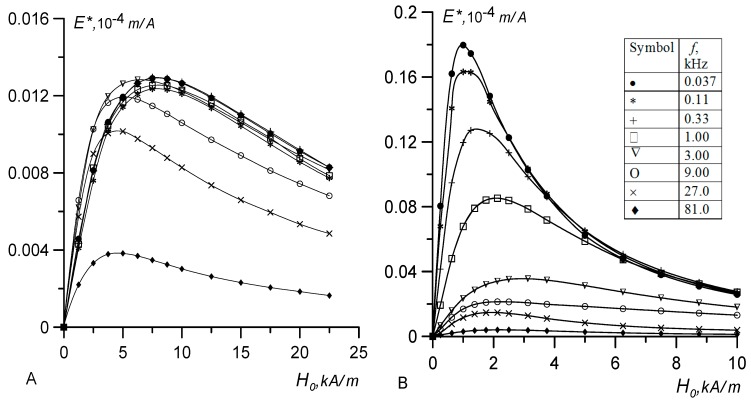
The normalized output signal versus the bias field strength for sample (**A**) No. 1 and (**B**) No. 2. Points indicate experimental data at different frequencies of the probe field; solid lines indicate spline smoothing.

**Figure 6 nanomaterials-09-01711-f006:**
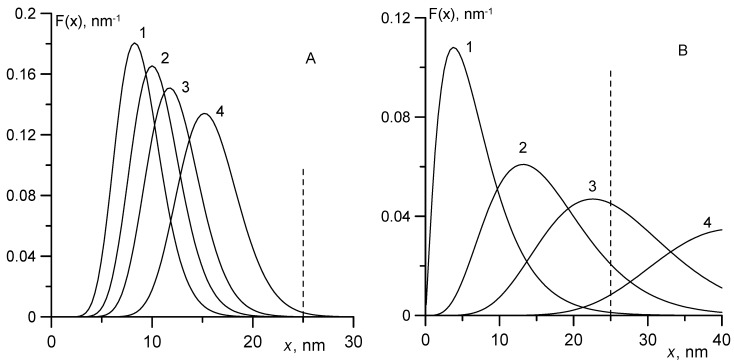
Γ-distribution of particle sizes (curve 1); the relative contribution of particles to the 3rd- <*x*^3^> (curve 2), 6th- <*x*^6^> (curve 3), and 12th-order moment <*x*^12^> (curve 4). The dashed line corresponds to the maximum possible diameter of magnetite particles in stable colloids. (**A**) sample No. 1; (**B**) sample No. 2.

**Table 1 nanomaterials-09-01711-t001:** Parameters of base magnetic fluids.

Physical Property	FM1	FM2
*T* (K)	291	286
*M_∞_* (kA/m)	66.5	13.3
*χ* _0_	4.40	4.86
*χ* _L_	2.39	2.56
<m> (10^−19^ A m^2^)	2.08	2.31
<m^2^> (10^−38^ A^2^ m^4^)	7.16	41.6
<*x*>, nm	8.82	6.92
*ρ* (g/cm^3^)	1.597	0.984
*φ* _ρ_	0.183	0.045
*δ* _x_	0.26	0.67

**Table 2 nanomaterials-09-01711-t002:** Magnetic properties of the samples used in the crossed fields experiments.

Physical Property	Sample 1	Sample 2
*T*(K)	297	298
*M_∞_* (kA/m)	7.9	4.4
*χ*	0.31	1.07
*χ* _L_	0.29	0.84
*λ*	0.6	2.1

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
