# Peer review of "Dynamics of Magnetic Fluids in Crossed DC and AC Magnetic Fields"

_nanomaterials, 2019, doi:10.3390/nano9121711_

Round 1

Reviewer 1 Report

This is an interesting contribution and worth publishing in this journal. The equations seem correct. The measured results are interpret appropriately.

Minor comments:

Almost half of the references cited are from the authors themselves. The authors not the only ones working on this topic. Please revise this carefully.

There is a question mark in the subscript of mu in equation 3.

Author Response

Response to reviewer (I)

Thank you very much for your interest in our manuscript and useful remarks. Your remarks and our answers are given below.

Almost half of the references cited are from the authors themselves. The authors not the only ones working on this topic. Please revise this carefully.

 Answer: We reduced to a minimum the number of references to our own works.

There is a question mark in the subscript of mu in equation 3.

Answer: The misprint is corrected.

Thank you again for your useful comments.

Reviewer 2 Report

In their manuscript “Dynamics of magnetic fluids in crossed magnetic fields” Pshenichnikov et al describe a Langevin-like model to describe the response of magnetic fluids in crossed DC/AC magnetic fields. The theoretical results are compared to experimental data obtained using two types of ferrofluids. In the experimental realization, the electromotive force (voltage) induced in a small probe coil is measured at the frequency two times higher than that of the applied orthogonal AC field. The authors show that the response is proportional to the 12th power of the particle size leading to very high sensitivity of the technique to the largest particles present in the tail of the particle size distribution. This is unfortunate since in most applications of ferrofluids lower-order momenta of the particle distribution function are of interest. Still, the research is well performed and the results are interesting. I, therefore, vote for publication of this work provided that the authors consider the following comments.

The usage of the English language is far from optimal. Just a couple of examples. Line 10: the abstract starts with “The article derived…”. I guess that the authors did that. Line 47: stratification -> separation(?). “…into highly and low concentrated phases” is not proper. Line 61: “As was shown in…” must be either “As shown in…” or “As it was shown in…”. Add comma before “the output signal” on the next line. What is “the density of size distribution” on line 72? Moreover, the usage of indefinite and definite articles is often incorrect.

How about adding DC/AC (or, DC and AC) to the title?

Abbreviation emf is not introduced.

Line 36: What is ferromagnetic resonance? Give a reference and mention the order-of-magnitude of the characteristic frequencies.

Line 50: The text refers explicitly to Figure 1B. Should not it be Figure 1 instead (or, Figure 1 A and B)?

The rightmost part of Eq. (3): why there is a small question mark hidden behind the equation?

Line 132: Should not letter a be replaced by alpha in the leftmost part of Eq. (8)?

The caption of Table 1 does not introduce the displayed quantities. The text right above the table does not help much: particle number density is announced in the text but is not shown in the table. Is chi shown in the table the same as chi_0 in Eq. (9)? Does g/sm^3 stand for g/cm^3? Is delta the same as delta_x?

Line 306: the text “the maximum on (‘of’ perhaps?) the chi’’ = f(omega) – curve” is not clear. I cannot make any sense out of the equality sign. Is f(omega) simply = 2pi*omega? Isn’t it too trivial to introduce it as a new function?

Who is A.O. in the Author Contribution section, who performed “formal analysis”(?)?

The reference list is rather long, to a large extent due to adding many self-citations. Is this a way to boost your own citation score or are all these publications essential for the understanding of this manuscript?

Author Response

Response to reviewer (II)

Thank you very much for your interest in our manuscript and useful remarks. Your remarks and our answers are given below.

The usage of the English language is far from optimal. Just a couple of examples. Line 10: the abstract starts with “The article derived…”. I guess that the authors did that. Line 47: stratification -> separation(?). “…into highly and low concentrated phases” is not proper. Line 61: “As was shown in…” must be either “As shown in…” or “As it was shown in…”. Add comma before “the output signal” on the next line. What is “the density of size distribution” on line 72? Moreover, the usage of indefinite and definite articles is often incorrect.

Answer: Thank you for helpful comments. The appropriate corrections are made. Manuscript "Dynamics of magnetic fluids in crossed magnetic DC and AC fields" has undergone English language editing by MDPI.

How about adding DC/AC (or, DC and AC) to the title?

Answer: We have accepted your suggestion.

Abbreviation emf is not introduced.

Answer: Abbreviation emf is introduced.

Line 36: What is ferromagnetic resonance? Give a reference and mention the order-of-magnitude of the characteristic frequencies.

Answer: We added references [4, 5] and introduced the characteristic frequency of ferromagnetic resonance equal to 1010 Hz.

Line 50: The text refers explicitly to Figure 1B. Should not it be Figure 1 instead (or, Figure 1 A and B)?

Answer: That’s right. Thank you for this remark.

The rightmost part of Eq. (3): why there is a small question mark hidden behind the equation?

Answer: The misprint is corrected.

Line 132: Should not letter a be replaced by alpha in the leftmost part of Eq. (8)?

Answer: That’s true. The error is corrected.

The caption of Table 1 does not introduce the displayed quantities. The text right above the table does not help much: particle number density is announced in the text but is not shown in the table. Is chi shown in the table the same as chi_0 in Eq. (9)? Does g/sm^3 stand for g/cm^3? Is delta the same as delta_x?

Answer: The misprints in Table 1 and the text right above the table are corrected. Thank you for these very useful remarks.

Line 306: the text “the maximum on (‘of’ perhaps?) the chi’’ = f(omega) – curve” is not clear. I cannot make any sense out of the equality sign. Is f(omega) simply = 2pi*omega? Isn’t it too trivial to introduce it as a new function?

Answer: The improper phrase is improved.

Who is A.O. in the Author Contribution section, who performed “formal analysis”(?)?

Answer: The misprint is corrected. A.O.is replaced by A.I.

Thank you again for your useful comments.